# Designing a Conditional Prior Distribution for Flow-Based Generative Models

## Abstract

Flow-based generative models have recently shown impressive performance for conditional generation tasks, such as text-to-image generation. However, current methods transform a general noise distribution to a specific mode of the target data distribution. As such, every point in the initial source distribution can be mapped to every point in the target distribution, resulting in a long average path. To this end, in this work, we tap into a non-utilized property of conditional flow-based models: the ability to design a non-trivial prior distribution. Given an input condition, such as a text prompt, we first map it to a point lying in data space, representing an "average" data point of the minimal average distance to all data points of the same conditional mode (e.g., class). We then utilize the flow matching formulation to map samples from a Gaussian centered around this point to the conditional target distribution. Experimentally, our method significantly improves training times and generation quality (FID, KID and CLIP alignment scores) compared to baselines, producing high quality samples using smaller number of sampling steps.

## 1 Introduction

Conditional generative models are of significant importance for many scientific and industrial applications. Of these, the class of flow-based models and score-based diffusion models has recently shown a particularly impressive performance (Lipman et al., 2022; Esser et al., 2024; Dhariwal & Nichol, 2021; Ho & Salimans, 2022). Although impressive, current methods suffer from long training and sampling times. To this end, in this work, we tap into a non-utilized property of conditional flow-based models: the ability to design a non-trivial prior distribution for conditional flow models based on the input condition. In particular, for class-conditional generation and text-to-image generation tasks, we propose a *robust* method for constructing a conditional flow-based generative model using an informative condition-specific prior distribution fitted to the conditional modes (e.g., classes) of the target distribution. By better modeling the prior distribution, we aim to improve the efficiency, both at training and at inference, of conditional generation via flow matching, thus achieving superior results with fewer sampling steps.

Given an input variable (e.g., a class or text prompt), current flow-based and score-based diffusion models combine the input condition with intermediate (possibly latent) representations in a learnable manner. However, crucially, these models are still trained to transform a generic unimodal noise distribution to the different modes of the target data distribution. In some formulations, such as score-based diffusion (Ho et al., 2020; Song et al., 2020; Sohl-Dickstein et al., 2015), the use of the source Gaussian density is intrinsically connected to the process of constructing the transformation. In others, such as flow matching (Lipman et al., 2022; Liu et al., 2022; Albergo & Vanden-Eijnden, 2022), a Gaussian prior is not required, but is often chosen as a default for convenience. Consequently, in this setting, the prior distribution bears little or no resemblance to the target, and hence every point in the initial source distribution can be mapped to every point in every mode in the target distribution, corresponding to a given condition. This means that the average distance between pairs of source-target points is fairly large.

In the unconditional setting, recent works (Pooladian et al., 2023; Tong et al., 2023), show that starting from a source (noise) data point that is close to the target data sample, during training, results in straighter probability flows, fewer sampling steps at test time, and faster training time. This is in

comparison to the non-specific random pairing between the distributions typically used for training denoising-based models. This suggests that finding a strategy to minimize the average distance between source and target points could result in a similar benefit. Our work aims to construct such a source distribution by leveraging the input condition.

We, therefore, propose a novel paradigm for designing an informative condition-specific prior distribution for a flow-based conditional generative model. While in this work, we choose to work on *flow matching*, our approach can also be incorporated in other generative models, supporting arbitrary prior distributions. In the first step, we embed the input condition $c$ to a point $x_c$ lying in the data space (which can be a latent representation). For a discrete set of classes, this is done by averaging training samples corresponding to a given class $c$ in the data space. In the continuous case, such as text-to-image, we first choose a meaningful embedding for the input condition $c$. For arbitrary texts, for instance, we choose the pretrained CLIP Radford et al. (2021) representation space. Given a training $x_c$ samples and the corresponding CLIP embedding $e_c$, we train a deterministic mapper function that projects $e_c$ to $x_c$ lying in data space. This results in an "average" data point corresponding to all samples $x$ with the same CLIP embedding $e_c$. To enable

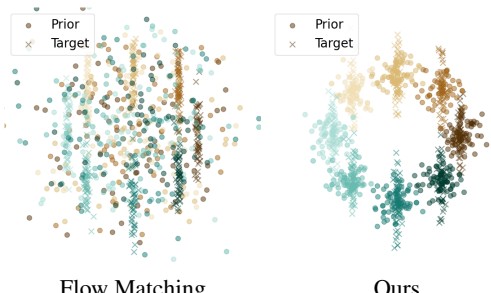

Flow Matching    Ours

Figure 1: **An illustration of our approach.** The LHS illustrates the standard flow matching, where every sample in the source Gaussian distribution (shown as a circular point) can be mapped to every sample in the conditional target mode (shown as a cross point), where each class samples are shown in a different color. In contrast, our method, shown on the RHS, constructs a class conditional GMM as a source prior distribution. Each sample in the source distribution is, on average, closer to its corresponding sample in the target mode.

stochastic mapping, we then map samples from a Gaussian centered on $x_c$ to the conditional target distribution $\rho_0(x_0|c)$. In the discrete case the covariance is estimated directly from class dependent training data, while for the continuous setting, it is fixed as a hyperparameter. An illustration of our approach, for a simple setting consisting of six Gaussians, each representing a different class, is shown in Fig. 1.

To validate our approach, we first formulate flow matching from the conditional prior distribution and show that our formulation results in a low truncation error. Next, we consider a toy setting with a known analytical target distribtuion and illustrate our method's advantage in efficiency and quality of generation. For real-world datasets, we consider both the MS-COCO (text-to-image generation) and ImageNet-64 datasets (class conditioned generation). Compared to other standard flow-based and diffusion (DDPM) based models, our approach allows for faster training and sampling, as well as for a significantly improved generated image quality and diversity, evaluated using FID and KID, and alignment to the input text, evaluated using CLIP score.

## 2 RELATED WORK

**Conditional Generation.** Conditional generative models have been modeled through a spectrum of generative architectures, including Generative Adversarial Networks (GANs) (Mirza & Osindero, 2014) and Variational Autoencoders (VAEs) (Sohn et al., 2015). Recently, innovations in diffusion models have enabled unprecedented performance in conditional generation, including text-to-image synthesis (Nichol et al., 2022; Ramesh et al., 2022; Saharia et al., 2024), text-to-video translation (Ho et al., 2022; Singer et al., 2023; Liu et al., 2024b), and text-to-speech conversion (Kong et al., 2021; Kim et al., 2021; Chen et al., 2020; Popov et al., 2021). Traditionally, conditional information is integrated into these networks via residual connections or cross-attention (Rombach et al., 2022). However, to the best of our knowledge, the literature lacks exploration of incorporating conditional information through the prior distribution of these models.

**Flow-based Models.** Continuous Normalizing Flows (CNFs) (Chen et al., 2019) emerged as a novel paradigm in generative modeling, offering a continuous-time extension to the discrete Normalizing Flows (NF) framework (Kobyzev et al., 2020; Papamakarios et al., 2021). Recently, Flow

Matching (Lipman et al., 2022; Liu et al., 2022; Albergo & Vanden-Eijnden, 2022) has been introduced as a simulation-free alternative for training CNFs. In scenarios involving conditional data (e.g., in text-to-image generation), conditioning is applied similarly to diffusion models, often through cross attention between the input condition and latent features. Typically, the source distribution remains unimodal, like a standard Gaussian (Liu et al., 2024a). In contrast, our approach derives a prior distribution that is dependent on the input condition.

**Informative Prior Design.** In the context of score-based models and flow matching, several works designed informative priors. For score-based diffusion, Lee et al. (2021) has introduced an approach of formulating the diffusion process using a non-standard Gaussian, where the Gaussian's statistics are determined by the conditional distribution statistics. However, this approach is constrained by the use of a Gaussian prior, which limits its overall flexibility. Recently, (Pooladian et al., 2023; Tong et al., 2023) constructed a prior distribution by utilizing the dynamic optimal transport (OT) formulation across mini-batches during training. Despite impressive capabilities such as efficient sampling (minimizing trajectory intersections), they suffer from several drawbacks: (i). Highly expensive training. Computing the optimal transport solution requires quadratic time and memory, which is not applicable to large mini-batches and high dimensional data. (ii). When dealing with high dimensional data, the effectiveness of this formulation decreases dramatically. An increase in performance requires an exponential increase in batch-size in relation to data dimension. Our approach avoids these limitations by leveraging the conditioning variable of the data distribution.

## 3 PRELIMINARIES

We begin by introducing Continuous Normalizing Flow Chen et al. (2019) in Sec. 3.1 and Flow Matching Lipman et al. (2022) in Sec. 3.2. This will motivate our approach, detailed in Sec. 4, which defines an informative conditional prior distribution on a conditional flow generative model.

### 3.1 CONTINUOUS NORMALIZING FLOW

A probability density function over a manifold $\mathcal{M}$ is a continuous non-negative function $\rho : \mathcal{M} \to \mathbb{R}_+$ such that $\int \rho(x)dx = 1$. We set $\mathcal{P}$ to be the space of such probability densities on $\mathcal{M}$. A *probability path* $\rho_t : [0,1] \to \mathcal{P}$ is a curve in probability space connecting two densities $\rho_0, \rho_1 \in \mathcal{P}$ at endpoints $t = 0, t = 1$. A *flow* $\psi_t : [0,1] \times \mathcal{M} \to \mathcal{M}$ is a time-dependent diffeomorphism defined to be the solution to the Ordinary Differential Equation (ODE):

$$\frac{d}{dt}\psi_t(x) = u_t(\psi_t(x)), \quad \psi_0(x) = x \tag{1}$$

subject to initial conditions where $u_t : [0,1] \times \mathcal{M} \to \mathcal{T}\mathcal{M}$ is a time-dependent smooth vector field on the collection of all tangent planes on the manifold $\mathcal{T}\mathcal{M}$ (*tangent bundle*). A flow $\psi_t$ is said to generate a probability path $\rho_t$ if it 'pushes' $\rho_0$ forward to $\rho_1$ following the time-dependent vector field $u_t$. The path is denoted by:

$$\rho_t = [\psi_t]_{\#}\rho_0 := \rho_0(\psi_t^{-1}(x)) \det \left| \frac{d\psi_t^{-1}}{dx}(x) \right| \tag{2}$$

where $\#$ is the standard push-forward operation. Previously, (Chen et al., 2019) proposed to model the flow $\psi_t$ implicitly by parameterizing the vector field $u_t$ using a neural network, to produce $\rho_t$, in a method called *Continuous Normalizing Flow* (CNF).

### 3.2 FLOW MATCHING

Flow Matching (FM) (Lipman et al., 2022) is a simulation-free method for training CNFs that avoids likelihood computation during training, which can be expensive and unstable. It does so by fitting a vector field $v_t^\theta$ with parameters $\theta$ and regressing vector fields $u_t$ that are known *a priori* to generate a probability path $\rho_t \in \mathcal{P}$ satisfying the boundary conditions:

$$\rho_0 = p, \quad \rho_1 = q \tag{3}$$

Note that $u_t$ is generally intractable. However, a key insight of Lipman et al. (2022), is that this field can be constructed based on conditional vector fields $u_t(x|x_1)$ that generate conditional probability

paths $\rho_t(x|x_1)$. The push-forward of the conditional flow $\psi_t(x|x_1)$, start at any $\rho_t$ and concentrate the density around $x = x_1 \in \mathcal{M}$ at $t = 1$. Marginalizing over the target distribution $q$ recovers the unconditional probability path and unconditional vector field:

$$\rho_t(x) = \int_{\mathcal{M}} \rho_t(x|x_1)q(x_1)dx_1 \tag{4}$$

$$u_t(x) = \int_{\mathcal{M}} u_t(x|x_1)\frac{\rho_t(x|x_1)q(x_1)}{\rho_t(x)}dx_1 \tag{5}$$

This vector field can be matched by a parameterized vector field $v_\theta$ using the following objective

$$\mathcal{L}_{\text{cfm}}(\theta) = \mathbb{E}_{t,q(x_1),\rho_t(x|x_1)}\|v_\theta(t,x) - u_t(x|x_1)\|^2, \quad t \sim \mathcal{U}(0,1). \tag{6}$$

where $\|\cdot\|$ is a norm on $\mathcal{T}\mathcal{M}$. One particular choice of a conditional probability path $\rho_t(x|x_1)$ is to use the flow corresponding the optimal transport displacement interpolant (McCann, 1997) between Gaussian distributions. Specifically, in the context of the conditional probability path, $\rho_0(x|x_1)$ is the standard Gaussian, a common convention in generative modeling, and $\rho_1(x|x_1)$ is a small Gaussian centered around $x_1$. The conditional flow interpolating these distributions is given by:

$$x_t = \psi_t(x|x_1) = (1-t)x_0 + tx_1. \tag{7}$$

which results in the following conditional vector field:

$$u_t(x|x_1) = \frac{x_1 - x}{1 - t} \tag{8}$$

which is marginalized in Eq. 6. Substituting Eq. 7 to Eq. 8, one can also express the value of this vector field using a simpler expression:

$$u_t(x_t|x_1) = x_1 - x_0 \tag{9}$$

**Conditional Generation via Flow Matching.** Flow matching (FM) has been extended to conditional generative modeling in several works (Zheng et al., 2023; Dao et al., 2023; Atanackovic et al., 2024; Isobe et al., 2024). In contrast to the original FM formulation, given in Eq. 8, one first samples a condition $c$. One then produces samples from $p_t(x|c)$ by passing $c$ as input to the parametric vector field $v_\theta$. The *Conditional Generative Flow Matching* (CGFM) objective is:

$$\mathcal{L}_{\text{cgfm}}(\theta) = \mathbb{E}_{t,q(x_1,c),\rho_t(x|x_1)}\|v_\theta(t,c,x) - u_t(x|x_1)\|^2, \quad t \sim \mathcal{U}(0,1). \tag{10}$$

In practice, $c$ is incorporated by embedding it into some representation space and then using cross-attention between this embedding and the features of $v_\theta$ as in Rombach et al. (2022).

**Flow Matching with Joint Distributions.** While Lipman et al. (2022) considered the setting of independently sampled $x_0$ and $x_1$, recently, Pooladian et al. (2023); Tong et al. (2023) generalized the FM framework to an arbitrary joint distribution of $\rho(x_0, x_1)$ in the unconditional generation setting. This construction satisfies the following marginal constraints, i.e.

$$\int \rho(x_0, x_1)dx_1 = q(x_0), \quad \int \rho(x_0, x_1)dx_0 = q(x_1) \tag{11}$$

Pooladian et al. (2023) proposed modifying the conditional probability path construction so at $t = 0$:

$$\rho_0(x_0|x_1) = p(x_0|x_1) \tag{12}$$

where $p(x_0|x_1)$ is the conditional distribution $\frac{\rho(x_0,x_1)}{q(x1)}$. The proposed *Joint Conditional Flow Matching* (JCFM) objective is:

$$\mathcal{L}_{\text{jcfm}}(\theta) = \mathbb{E}_{t,\rho(x_0,x_1)}\|v_\theta(t,x) - u_t(x|x_1)\|^2, \quad t \sim \mathcal{U}(0,1). \tag{13}$$

## 4 METHOD

Given a set $\{x_{1_i}, c_i\}_{i=1}^m$ of input samples and their corresponding conditioning states, our goal is to construct a flow-matching model that samples from $q(x_1|c)$ using flows that start from our conditional prior distribution (CPD).

### 4.1 FLOW MATCHING FROM CONDITIONAL PRIOR DISTRIBUTION

We generalize the framework of Sec. 3.2 to a construction that uses an arbitrary conditional joint distribution of $\rho(x_0, x_1, c)$ which satisfy the marginal constraints:

$$\int \rho(x_0, x_1, c)dx_0 = q(x_1, c) \quad \int \rho(x_0, x_1, c)dx_1 dc = p(x_0) \tag{14}$$

Then, building on flow matching, we propose to modify the conditional probability path so that at $t = 0$, we define:

$$\rho_0(x_0|x_1, c) = p(x_0|x_1, c) \tag{15}$$

where $p(x_0|x_1, c)$ is the conditional distribution $\frac{\rho(x_0, x_1, c)}{q(x_1, c)}$. Using this construction, we satisfy the boundary condition of Eq. 3:

$$\rho_0(x_0) = \int \rho_0(x_0|x_1, c)q(x_1, c)dx_1 dc = \int p(x_0|x_1, c)dx_1 dc = p(x_0) \tag{16}$$

Note that the conditional probability path $\rho_t(x|x_1, c)$ does not need to be explicitly formulated. Instead, only its corresponding conditional vector field $u_t(x|x_1, c)$ needs to be defined such that points $x_0$ drawn from the conditional prior distribution $\rho_0(x_0|x_1, c)$, reach $x_1$ at $t = 1$, i.e., reach distribution $\rho_1(x|x_1, c) = \delta(x - x_1)$. We thus purpose the following *Conditional Generation Joint FM* objective:

$$\mathcal{L}_{\text{cgjfm}}(\theta) = \mathbb{E}_{t, q(x_0, x_1, c)} \|v_\theta(t, x, c) - u_t(x|x_1, c)\|^2, \quad t \sim \mathcal{U}(0, 1). \tag{17}$$

where $x = \psi_t(x_0|x_1, c)$. Training only involves sampling from $q(x_0, x_1, c)$ and does not require explicitly defining the densities $q(x_0, x_1, c)$ and $\rho_t(x|x_1, c)$. We note that this objective is reduced to the CGFM objective Eq. 10 when $q(x_0, x_1, c) = q(x_1, c)p(x_0)$.

### 4.2 CONDITIONAL PRIOR DISTRIBUTION

We now describe our choice of a condition-specific prior distribution. We design a condition-specific prior distribution based on a *Gaussian Mixture Model* (GMM) where each mode of the mixture is correlated to a specific conditional distribution $p(x_1|c)$. Specifically, we choose the prior distribution to be the following:

$$p_0 = \text{GMM}(\mathcal{N}(\mu_i, \Sigma_i)_{i=1}^n, \pi) \tag{18}$$

where $\pi \in \mathbb{R}^n$ is a probability vector associated with the number of conditions $n$ (could be $\infty$) and $\mu_i, \Sigma_i$ are parameters determined by the conditional distribution $q(x_1|c_i)$ statistics, *i.e.*

$$\mu_i = \mathbb{E}[x_1|c_i], \quad \Sigma_i = \text{cov}[x_1|c_i] \tag{19}$$

To sample from the marginal distribution $p(x_0|x_1, c_i)$, we sample from the cluster $\mathcal{N}(\mu_i, \Sigma_i)$ associated with the condition $c_i$.

**Obtaining a Lower Global Truncation Error.** CPD fits a GMM to the data distribution in a favorable setting, where the association between samples and clusters is given. In this process, we fit a dedicated Gaussian distribution to data points with the same condition. If the latter are close to being unimodal, this approximation is expected be tight, in terms of the average distances between samples from the condition data mode and the fitted Gaussian. Tab. 1 provides the average distances between pairs of samples from the prior and data distributions of CondOT Lipman et al. (2022), BatchOT Pooladian et al. (2023) and our CPD over the ImageNet-64 Deng et al. (2009) and MS-COCO Lin et al. (2014) datasets. As expected, BatchOT which minimizes this exact measure within mini-batches, obtains better scores than the naïve pairing used in CondOT, while our CPD, which approximates

|         | ImageNet-64 | MS-COCO |
|---------|-------------|---------|
| CondOT  | 640         | 630     |
| BatchOT | 632         | 604     |
| Ours    | **570**     | **510** |

Table 1: Average distances between pairs of samples from the prior and data distributions on the ImageNet64 and MS-COCO datasets across baselines.

the data using a GMM exploits the conditioning available in these datasets, and obtains considerably lower average distances.

As noted in Pooladian et al. (2023), shorter distances are generally associated with straighter flow trajectories, more efficient sampling and lower training time. We want to substantiate this claim from the viewpoint of cumulative errors in numerical integration. Sampling from flow-based models consists of solving a time-dependent ODE of the form $\dot{x}_t = u_t(x_t)$, where $u_t$ is the velocity field. This equation is solved by the following integral $x_t = \int_0^t u_s(x_s)ds$, where the initial condition $x_0$ is sampled from the prior distribution. Numerical integration over discrete time steps accumulate an error at each step $n$ which is known as the *local truncation error* $\tau_n$, which accumulates into what is know as the *global truncation error* $e_n$. This error is bounded by Süli & Mayers (2003)

$$|e_n| \leq \frac{max_j \tau_j}{hL}\left(e^{L(t_n - t_0)} - 1\right) \tag{20}$$

where $h$ is the step size and $L$ is the Lipschitz constant of the velocity $u_t$. According to the above, the distance between the endpoints of a path $\Delta = |x_1 - x_0|$ is given by $|\int_0^1 u_s(x_s)ds|$ which can be interpreted as the magnitude of the average velocity along the path $x_t$. Hence, the longer the path $\Delta$ is, the larger the integrated flow vector field $u_t$ is. For example, if we scale a path uniformly by a factor $C > 1$, i.e., $x_t \to C(x_t)$, we get, $\frac{d}{dt}C(x_t) = C(u_t)$ in which case the Lipschitz constant $L$ is also multiplied by $C$.

By shortening the distance between the prior and and data distribution, as our CPD does, we lower the integration errors which permits the use of coarser integration steps, which in turn yield smaller global errors. Thus, our construction allows for smaller number of integration steps during sampling.

### 4.2.1 Construction

Next, we explain how we construct $p_0$ (Eq. 18) for both the discrete case (e.g., class conditional generation) and continuous case (e.g., text conditional generation).

**Discrete Condition.** In the setup of a discrete conditional generation task, we are given a data $\{x_{1_i}, c_i\}_{i=1}^m$ where there are a finite set of conditions $c_i$. We approximate the statistics of Eq. 19 directly using the training data statistics. That is, we compute the mean and covariance matrix of each class (potentially in some latent represntation of a pretrained auto-encoder). Since the classes at inference time are the same as in training, we use the same statistics at inference.

**Continuous Condition.** While in the discrete case we can directly approximate the statistics in Eq. 19 from the training data, in the continuous case (*e.g.* text-conditional) we need to find those statistics also for conditions that were not seen during training. To this end, we first consider a joint representation space for training samples $\{x_{1_i}, c_i\}_{i=1}^m$, which represents the semantic distances between the conditions $c_i$ and the samples $x_{1_i}$. In the setting where $c_i$ is text, we choose a pretrained CLIP embedding. $c_i$ is then mapped to this representation space, and then mapped to the data space (which could be a latent representation of an auto-encoder), using a learned mapper $\mathcal{P}_\theta$. Specifically, $\mathcal{P}_\theta$ is trained to minimize the objective:

$$\mathcal{L}_{\text{prior}}(\theta) = \mathbb{E}_{q(x_1, c)}\|\mathcal{P}_\theta(E(c)) - x_1\|_2^2. \tag{21}$$

where $E$ is the pre-trained mapping to the joint condition-sample space (e.g. CLIP). $\mathcal{P}_\theta$ can be seen as approximating $\mathbb{E}[x_1|c]$, which is used as the mean for the condition specific Gaussian. At inference, where new conditions (e.g., texts) may appear, we first encode the condition $c_i$ to the joint representation space (e.g., CLIP) followed by $\mathcal{P}_\theta$. This mapping provides us with the center $\mu_i$ of each Gaussian. We also define $\Sigma_i = \sigma_i^2 I$ where $\sigma_i$ is a hyper-parameter, ablated in Sec. 5.2.1

### 4.3 Training and Inference

Given the prior $p_0$ (either using the data statistics or by training $\mathcal{P}_\theta$), for each condition $c$, we have its associated Gaussian parameters $\mu_c$ and $\Sigma_c$. The map $\psi_t(x|x_1, c)$ must be defined in order to minimize Eq. 17 above. This corresponds to the interpolating maps between this Gaussian at $t = 0$ and a small Gaussian around $x_1$ at $t = 1$, defined by:

$$\psi_t(x|x_1, c) = \sigma_t(x_1, c)x + \mu_t(x_1, c), \tag{22}$$

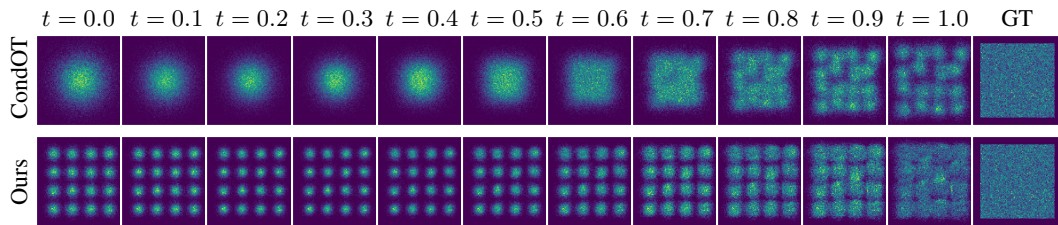

Figure 2: **Trajectory illustration.** A toy example illustrating the trajectory from the source to the target distribution for our method and conditional flow matching using optimal transport (CondOT).

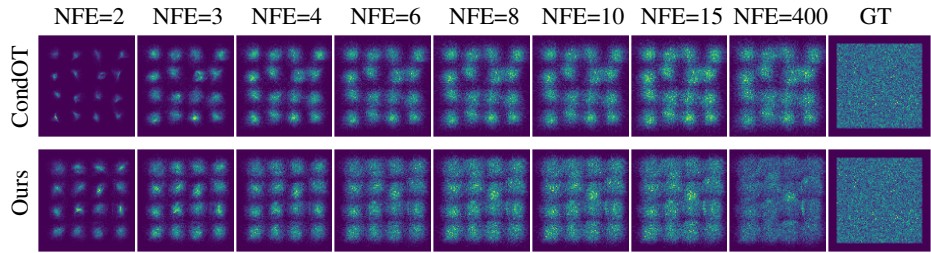

Figure 3: **NFE convergence illustration.** A toy example illustrating convergence to the target distribution at different NFEs, for our method, compared to class conditioned flow matching using optimal transport paths (CondOT).

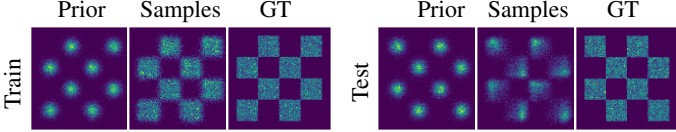

Figure 4: **Generalization illustration.** A toy example illustrating the generalization capabilities. LHS: Source prior and target samples for training classes RHS: As for LHS, but for test classes.

where

$$\sigma_t(x_1, c) = t(\sigma_{\min}I) + (1-t)\Sigma_c^{1/2}, \quad \text{and} \quad \mu_t(x_1, c) = tx_1 + (1-t)\mu_c. \tag{23}$$

This results in the following target flow vector field

$$u_t(x|x_1, c) = \frac{d}{dt}\psi_t(x|x_1, c) = \left(\sigma_{\min}I - \Sigma_c^{1/2}\right)x + x_1 - \mu_c. \tag{24}$$

During inference we are given a condition $c$ and want to sample from $q(x_1|c)$. Similarly to the training, we sample $x_0 \sim p(x_0|c)$ and solve the ODE

$$\frac{d}{dt}\psi_t(x) = v_\theta\left(t, \psi_t(x), c\right), \quad \psi_0(x) = x_0 \tag{25}$$

Our implementation details are provided in Appendix B

## 5 EXPERIMENTS

We begin by validating our approach on a 2D toy example. Subsequently, for two real-world datasets, we evaluate our approach on class-conditional and text-conditional image generation.

### 5.1 TOY EXAMPLE

We begin by considering the setting in which the prior distribution is a mixture of isotropic Gaussians (GMM), where each Gaussian's mean represents the center of a class (we set the standard deviation to 0.2). The target distribution consists of 2D squares with the same center as the Gaussian's mean in the source distribution and with a width and height of 0.2, representing a large square. We compare

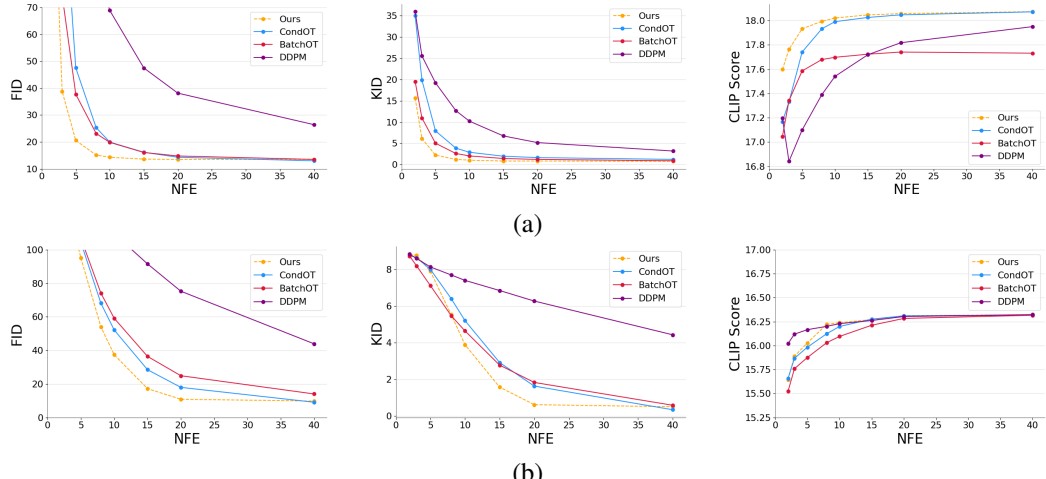

Figure 5: **Numerical evaluation.** (a) We compare our method to class conditional flow matching using optimal transport paths (CondOT), Pooladian et al. (2023) (BatchOT), and DDPM, on the ImageNet-64 dataset. We consider the FID score (LHS), KID score (Middle) and CLIP score (RHS). (b). As in (a) but for text-to-image generation on the MS-COCO dataset. As can be seen our method exhibit significant improvement per NFE, especially for low NFEs. For example, for 15 NFEs, on ImageNet-64 and MS-COCO we get **FID of 13.62** and **FID of 18.05** respectively, while baselines do not surpass FID of 16.10 and FID of 28.32 respectively for the same NFEs.

our method to class-conditional flow matching (with OT paths), where each conditional sample can be generated from each Gaussian in the prior distribution.

In Fig. 2, we consider the trajectory from the prior to the target distribution. By starting from a more informative conditional prior, our method converges more quickly and results in a better fitting of the target distribution. In Fig. 3, we consider the resulting samples for the different NFEs. NFE indicates the number of function evaluation required for an adaptive step solver to reach a pre-defined numerical tolerance. Specifically, we use `dopri5` sampler with `atol=rtol=1e-5` from the `torchdiffeq` (Chen, 2018) library. As can be seen, our method better aligns with the target distribution with fewer number of steps.

In Fig. 4, we consider the ability of our model to generalize to new classes not seen during training, akin to the setting of text-to-image generation. By training on only a subset of the classes our model exhibits generalization to new classes at test time

## 5.2 REAL WORLD SETTING

**Datasets and Latent Representation Space.** For the class-conditioned setting, we consider the ImageNet-64 dataset Deng et al. (2009), which includes more than 1.28M training images and 50k validation images, categorized into 1k object classes, all resized to $64 \times 64$ pixels. For the text-to-image setting, we consider the 2017 split of the MS-COCO dataset Lin et al. (2014), which consists of 330,000 images annotated with 80 object categories and over 2.5 million labeled instances. We use the standard split of 118k images for training, 5k for validation, and 41k for testing. We compute all our metrics on the ImageNet-64 validation set and the MS-COCO validation set. We perform flow matching in the latent representation of a pre-trained auto-encoder van den Oord et al. (2018).

### 5.2.1 QUANTITATIVE RESULTS

For a fair comparison, we evaluate our method in comparison to baselines using the same architecture, training scheme, and latent representation, as detailed above. We compare our method to standard class-conditioned or text-conditioned flow matching with OT paths Lipman et al. (2022) which we denote CondOT, where the source distribution is chosen to be a standard Gaussian. We also consider BatchOT Pooladian et al. (2023), which constructed a prior distribution by utilizing the dynamic optimal transport (OT) formulation across mini-batches during training. Lastly, we

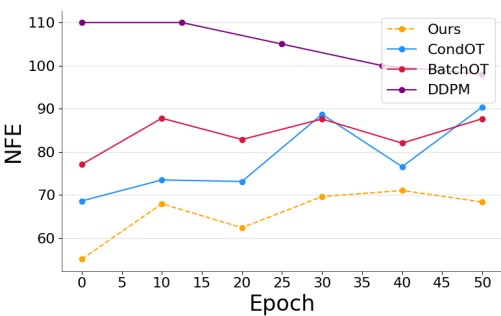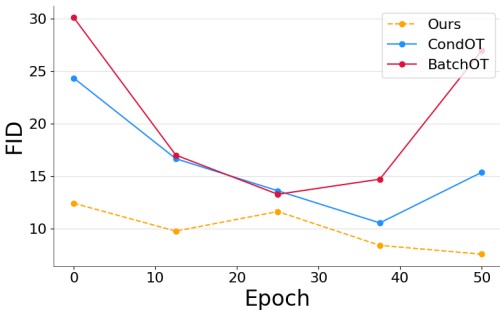

Figure 6: **Training time.** For a text-conditional model trained on MS-COCO, we consider the NFE per training epoch. We compare our method with text conditional flow matching using optimal transport paths (CondOT)Lipman et al. (2022), Pooladian et al. (2023), and DDPM. Note that DDPM had an FID value above 30 for all epochs so not shown on the LHS.

consider Denoising Diffusion Probabilistic Models (DDPM) Ho et al. (2020). To evaluate image quality, we consider the KID Bińkowski et al. (2021) and FID Heusel et al. (2018) scores commonly used in literature. We also consider the CLIP score to evaluate the alignment of generated images to the input text or class, using the standard setting, as in Hessel et al. (2022).

**Overall Performance.** We evaluate the FID, KID and CLIP similarity metrics for various NFE values (as defined above), which is indicative of the sampling speed. In Fig. 5(a) and Fig. 5(b), we perform this evaluation for our method and the baseline methods, for ImageNet-64 (class conditioned generation) and for MS-COCO (text-to-image generation), respectively. As can be seen, our method obtains superior results across all scores for both ImageNet-64 and MS-COCO. For ImageNet-64, already, at 15 NFEs our method achieves almost full convergence, whereas baseline methods achieve such convergence at a much higher NFEs. This is especially true for FID, where our method converges at 15 NFEs, and baseline methods only achieve such performance at 30 NFEs. A similar behavior occurs for MS-COCO at 20 NFEs.

|  | FID ↓ | KID↓ | CLIP↑ |
|---|---|---|---|
| $\sigma = 0.2$ | 23.55 | 2.88 | **16.12** |
| $\sigma = 0.5$ | 15.47 | 0.93 | 15.75 |
| $\sigma = 0.7$ | **7.55** | **0.61** | 15.85 |
| $\sigma = 1.0$ | 7.87 | 1.66 | 15.81 |
| w/o CLIP | 16.33 | 2.38 | 15.51 |

Table 2: **Ablation study.** We consider the model perfromance for different values of $\sigma$ (the standard deviation) as a hyperparameter for a model trained on MS-COCO. We also consider the case where our mapper $P_\theta$ takes as input a bag-of-words encoding instead of a CLIP.

**Training Convergence Speed.** By starting from our conditional prior distribution, training paths are on average shorter, and so our method should also converge more quickly at training. To evaluate this, in Fig. 6, we consider the NFE and FID obtained at each epoch, compared to baselines, for a model trained on MS-COCO. FID is computed using an Euler sampler with a constant number of function evaluations, NFE=20. Our method results in lower NFEs and superior FID, for every training epoch.

**Qualitative Results.** In Fig. 7, we provide a visualization of our results for a model trained on MS-COCO. We show, for four different text prompts: (a). The sample corresponding to the text in the conditional source distribution, which is used as the center of Gaussian corresponding to the text prompt. (b). Six randomly generated samples from the learned target distribution conditioned on the text prompt. As can be seen, the conditional source distribution samples resemble 'an average' image corresponding to the text, while generated samples display diversity and realism. In Appendix A, we also provide a diverse set of images generated by our method, in comparison to flow matching.

In Fig. 8, we consider, for a model trained on MS-COCO and a specific prompt, a visualization of our results for different NFEs, illustrating the sample quality for varying degrees of sampling times. As can be seen, our method already produces highly realistic results at NFE=15.

**Ablation Study.** In the continuous setting, as in MS-COCO, our method requires the choice of the hyperparameter $\sigma$, the standard deviation of each Gaussian. In Tab. 2, we report the FID, KID, and CLIP similarity values for different values of $\sigma$. As can be seen, our method results in best performance when $\sigma = 0.7$. We also consider the case where our mapper $\mathcal{P}_\theta$ takes as input a bag-of-words encoding instead of a CLIP encoding. As can be seen, performance drops significantly.

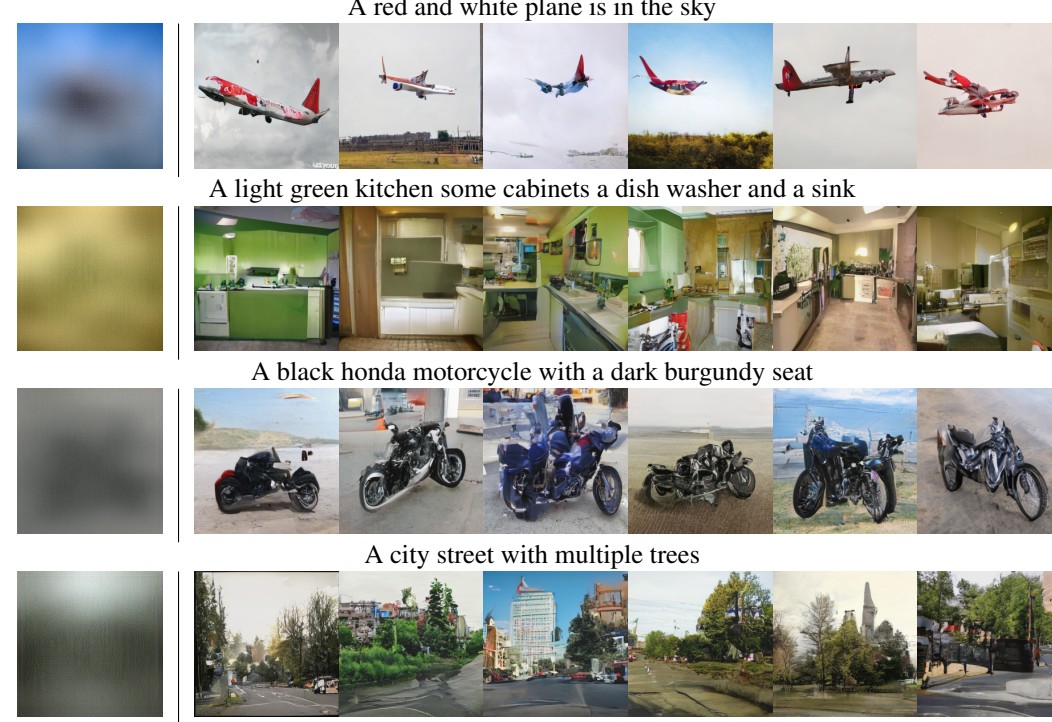

Figure 7: **A visualization of our results on MS-COCO.** We show, for four different text prompts: (a). The sample corresponding to the text in the conditional source distribution, which is used as the center of Gaussian corresponding to the text prompt (LHS) (b). Six randomly generated samples from the learned target distribution conditioned on the text prompt (RHS).

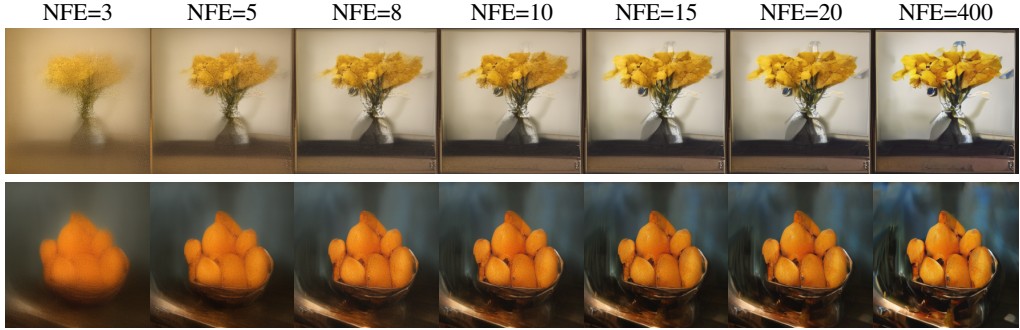

Figure 8: **A visualization of our results for different NFEs**. We consider a model trained on MS-COCO, and two different validation prompts: Top: "There are yellow flowers inside a vase", Bottom: "A bowl full of oranges".

## 6   CONCLUSION

In this work, we introduce a novel initialization for flow-based generative models using condition-specific priors, improving both training and inference efficiency. Our method allows for significantly shorter flow paths, reducing the global truncation error. Our approach achieves state-of-the-art performance on MS-COCO and ImageNet-64, surpassing baselines in FID, KID, and CLIP scores, particularly at lower NFEs. The flexibility of our method opens avenues for further exploration of other conditional initialization. While this work we assumed a GMM structure of the prior distribution, different structures can be explored. Furthermore, one could incorporate additional conditions such as segmentation maps or depth maps.

**Reproducibility Statement.** Experiments were conducted using 4 NVIDIA L40S GPUs. All code and scripts required to reproduce the experiments, including training, evaluation, and sampling, will be made fully available upon acceptance. In terms of data access, for the real-world image generation tasks, we use the publicly available ImageNet-64 and MS-COCO datasets.

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

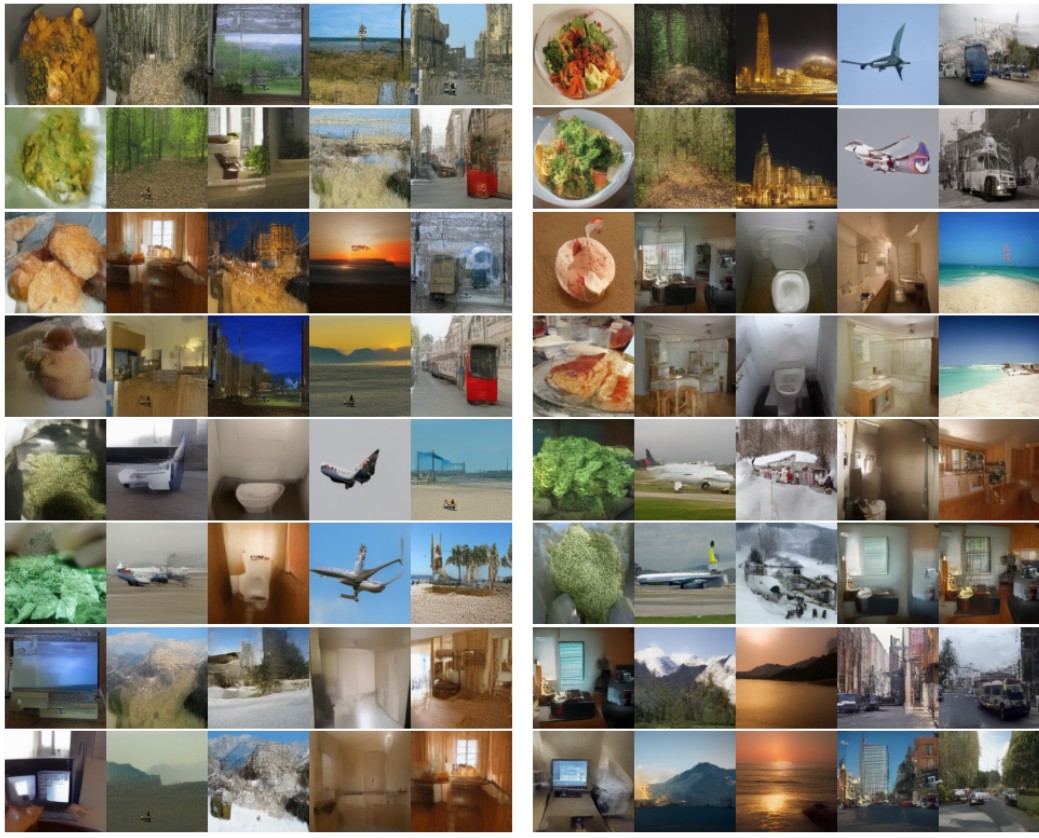

Flow Matching                                    Ours

Figure 9: Visual comparison of randomly generated samples for prompts from the MS-COCO validation set using our method, in comparison to flow matching, for a model trained on MS-COCO.

|                        | ImageNet-64 | MS-COCO |
|------------------------|-------------|---------|
| Dropout                | 0.0         | 0.0     |
| Effective Batch size   | 2048        | 128     |
| GPUs                   | 4           | 4       |
| Epochs                 | 100         | 50      |
| Learning Rate          | 1e-4        | 1e-4    |
| Learning Rate Scheduler| Constant    | Constant|

Table 3: Hyper-parameters used for training each model

## A  VISUAL RESULTS

In Fig. 9, we provide additional visual results for our method in comparison to standard flow matching for a model trained on MS-COCO.

## B  IMPLEMENTATION DETAILS

We report the hyper-parameters used in Table 3. All models were trained using the Adam optimizer Kingma & Ba (2017) with the following parameters: $\beta_1 = 0.9$, $\beta_2 = 0.999$, weight decay = 0.0, and $\epsilon = 1e-8$. All methods we trained (*i.e.* Ours, CondOT, BatchOT, DDPM) using identical architectures, specifically, the standard Unet Ronneberger et al. (2015) architecture from the `diffusers` von Platen et al. (2022) library with the same of parameters for the the same number of Epochs (see

Table 3 for details). For all methods and datasets, we utilize a pre-trained Auto-Encoder van den Oord et al. (2018) and perform the flow/diffusion in its latent space.

In the case of text-to-image generation, we encode the text prompt using a pre-trained CLIP network and pass to the velocity $v_\theta$ using the standard Unet condition mechanism. In the class-conditional setting, we create the prompt 'an image of a $\langle class \rangle$' and use it for the same conditioning scheme as in text conditional generation.

For the mapper $\mathcal{P}_\theta$ from Sec 4.2 we use a network consisting a linear layer and 2 ResNet blocks.

When using an adaptive step size sampler, we use `dopri5` with `atol=rtol=1e-5` from the `torchdiffeq` (Chen, 2018) library.

Regarding the toy example Sec. 5.1, we use a 4 layer MLP as the velocity $v_\theta$. In this setup, we incorporate the condition by using positional embedding Vaswani et al. (2023) on the mean of each conditional mode and pass it to the velocity $v_\theta$ by concatenating it to its input.

## C SOCIAL IMPACT STATEMENT

Conditional generative models, such as text-conditioned flow-based models, have a broad social impact on many applications, including content creation, advertising, and manufacturing. By improving their efficiency and accuracy, one can further enhance their applicability. However, their ability to generate photorealistic, targeted content also introduces risks, such as the potential for creating deep fakes, making it essential to ensure responsible usage and ethical safeguards in their deployment.

