# OpenReview forum: "Designing a Conditional Prior Distribution for Flow-Based Generative Models"
_ICLR.cc/2025/Conference — ICLR 2025 Conference Withdrawn Submission_

### Official Review · Reviewer_LWo5 · 2024-10-29

**Soundness:** 3
**Presentation:** 3
**Contribution:** 2
**Rating:** 3
**Confidence:** 4

**Summary:**

This paper proposes a method for conditional generation via flow matching, distinctively using a Gaussian Mixture Model (GMM) instead of a simple Gaussian as the prior distribution. This aims to reduce the distance between the prior and target distributions. Through truncation error analysis and toy experiments, the authors demonstrate that the method effectively minimizes this distance, improving the prior distribution. The authors apply the approach to class-conditional generative models on ImageNet-64 and text-to-image models on MS-COCO, showing promising results. However, GMM’s limitations and the diverse nature of text embeddings in text-to-image generation raise doubts about its applicability in this domain.

**Strengths:**

1. The paper flows seamlessly from the problem statement and proposed solution to hypothesis validation and experimental results. It clearly explains the importance of minimizing the prior-target distribution distance, presents an effective solution, and demonstrates the distance reduction experimentally. Additionally, when applied to generative models, FID, KID, and CLIP scores improve, showing a consistent and well-structured approach from start to finish.
2. In class-conditional image generation, the proposed method outperforms other techniques in metrics such as FID, KID, and CLIP Score.

**Weaknesses:**

1. The proposed solution appears overly simplistic and may be ineffective for text-to-image generation. As acknowledged in the paper, text space is continuous, making it questionable to model the prior distribution with a GMM. This issue likely explains why, although the method improves CLIP Score on ImageNet, it underperforms compared to baseline Stable Diffusion on MS-COCO, as shown in Figure 5. Even it is possible to decrease FID while increasing the CLIP Score as shown in Table 2, the performance gain is too minimal.
2. Moreover, MS-COCO is too simple a benchmark for evaluating text-to-image performance, given its low text diversity. Performance evaluation on more challenging benchmarks, such as PickScore or DrawBench datasets, would strengthen the claim.
3. Even with the use of a pretrained auto-encoder to convert images to latent space, it’s challenging to justify that the transformed latents adhere to a GMM prior. While the distance reduction relative to BatchOT and CondOT is noted, the GMM prior itself remains unconvincing.
4. Minor issue: According to ICLR formatting guidelines, table captions should be placed above the tables.

The main concern is that a GMM-based prior may be too simplistic to represent text-to-image space effectively. Demonstrating that FID and KID scores improve while maintaining or increasing text-image alignment on more complex benchmarks would improve the evaluation.
I will increase my score if my concerns are addressed.

**Questions:**

Why is the scale of the CLIP Score so small? I found it varies in the range of 16~18 but it is unusual.

---

### Official Review · Reviewer_4dtj · 2024-11-01

**Soundness:** 2
**Presentation:** 2
**Contribution:** 1
**Rating:** 3
**Confidence:** 4

**Summary:**

This research presents a design of a prior distribution for Flow matching with Gaussian Mixture Model, with means and variances  parametrized by conditional means and conditional variance.
The research showcases the efficacy of their design through experiments.

**Strengths:**

This research presents very thorough experiments on standard evaluations of generative models of Flow matching type, including NFE and CLIP score (In Image generation task).

**Weaknesses:**

- The reviewer is unfortunately a little skeptical regarding the extent of the novelty of the research, as its novelty winds down to, if the reviewer is not mistaken, the proposition of choosing Gaussian Mixture Distribution as the source distribution. The reviewer would like to note that the generative scheme proposed by Atong et al (Conditional Flow Matching) can be applied to "the generation of samples from $c$-conditioned distribution $\mu^c$   by training a "c-parametrized" vector field $v(t, x | c)$  through the loss $$E_{c, x_1^c, x_0^c} [ \| v(t, \psi_c(t) | c) -   (x_1^c - x_0^c) \|^2] $$
with $x_1^c  \sim \mu^c_1$, $x_0^c \sim \mu_0^c,  \psi_c(t) = t x_1^c - (1-t) x_0^c$, and that it is customary to choose   $\mu_0^c = N(E[x|c],  Var[x|c])$ in practice, instead of a non-informative, c-independent prior.  At the time of the generation, one may just integrate the ODE
$$\dot{x}^c(t) =  v(t, x^c(t) | c),   ~~~~   x^c(0) \sim \mu^c_0   $$
forward in time.
  This scheme is mathematically consistent as well, because it simply amounts to simultaneously solving $c$-parameterized set of continuity equations for $\{(\mu^c_t,  v(t, x_t | c) ) \}.$    Such a scheme appears, for example, in  Isobe et al (Extended Flow matching) as well, and the reviewer feels that this has been explored elsewhere as well.  Note that this scheme only differs from the proposed method only in that the latter chooses $\mu_0^c= \mu_0 = Mixture({N(E[x|c_i],  Var[x|c_i]})$.
In such a scheme, it is also critical that the regressors $m: c \to E[x|c]$,  $v: c \to Var[x|c]$  to be used in the way of $\mu_0^c= N(m(c),  v(c)) $ is trained so that they can well inter/extrapolate $E[x|c], Var[x|c]$ for $c$ not in training (corresponds to (21) in this research). The reviewer acknowledges that the paper partially mentions this matter,  but the reviewer also believes that it is particularly a nontrivial problem when $c$ is "not" dense everywhere.  That being said, the reviewer would like to know if the followings have been considered:
    - How does the current GMM construction of the prior compare against the usage of c-parametrized prior $\mu_0^c = N(E[x|c],  Var[x|c])$ for the generation of $\mu^c$ with $c$-parametrized vector field  $v(t, x | c)$?  Is there any merit in choosing common $\mu_0 = Mixture({N(E[x|c_i],  Var[x|c_i]})$ for the generation of $\mu^c$ for each $c$ ?
        - if the choosing of common $\mu_0$ is essential, what would be the mechanism behind it?  What are the situations in which the choosing of common $\mu_0$ would be beneficial?
    -  How does the current regression scheme ($P_\theta$) fares in various situations, such as with the presence of
        - condition c that lies far from the bulk of the conditions (outlying c)
        - condition c with small number of samples for $\mu^c$ (rare condition)?
    -  The Design of Source distribution as the one the reviewer presented above as well as the design of GMM distribution presented in this paper seem to come from the intuition that Flow matching performs better when the Wasserstein distance between the source distribution and the target distribution is smaller.  Has any theoretical investigation been done in this direction? How would the choice of isotropic Gaussian in GMM empirically fare against non-isotropic counterpart when each conditional distribution is highly non gaussian?

- While the reviewer very much values the amount of experiments done, for the reason the reviewer outlayed above, the reviewer feels that more theoretical / more ablative studies are required to substantiate the contribution.

**Questions:**

Please see the section above.

---

### Official Review · Reviewer_2aYP · 2024-11-01

**Soundness:** 3
**Presentation:** 2
**Contribution:** 3
**Rating:** 6
**Confidence:** 4

**Summary:**

Unlike other generative model families, flows are relatively unconstrained in the choice of the source distribution. This paper mainly poses that it should be possible to utilize this characteristic in a beneficial manner during conditional generative learning: by designing a condition-specific prior distribution, e.g. a mixture of Gaussians constructed from dataset statistics. By using the proposed method, the overall transport cost is reduced, which can result in straighter trajectories and better samples with fewer function evaluations.

**Strengths:**

## Strengths
1. The paper is well written and has sufficient clarity.
2. The overall idea of the paper is intuitive to grasp. Further, adopting the proposed approach should only need very minimal modification to regular conditional generative tasks---only a mean-variance calculation needs to be computed on the class-partitioned data, which is usually feasible.
3. The experiments show clear improvements over existing methods.

**Weaknesses:**

## Weaknesses
1. One of the key ideas of the paper is that shorter average distances yield straighter trajectories (e.g. L273), which is already discussed previously in the work of Pooladian et al. 2023 [w-a]. However, none of the toy examples (Fig 2, 3, 4) show the trajectories taken by the proposed approach. I suggest presenting some trajectory diagrams for these toy experiments (for instance, like the trajectories shown in the minibatch OT paper [w-b]).
2. One of the implicit assumptions of the paper is that in a class conditional dataset, each class is a mode, and is sufficiently disentangled from the other classes. Additionally, there is also an assumption of homogeneity in the data of each class, ignoring the possibility of internal modes in a class. It would be useful for the paper if analysis was provided on different types of *conditional data distributions*.
    - For example, consider a dataset from the [Datasaurus Dozen](https://jumpingrivers.github.io/datasauRus/), like VLines. Suppose line 1 and 3 (the odd lines) in VLines are one class (A), while the even 2 and 4 are another class (B). The mean of class A falls on samples from class B, and vice versa. Showing that the proposed approach still works better in this case than a standard normal would add to the strength of the paper. Or if not, it would be useful to know what properties of the target dataset are ill-suited for the proposed approach.
3. Rather than the irrelevant DDPM, the comparison should include RectifiedFlow, (e.g. 2-RF) [w-c] that is well established to have straight paths and low NFEs.
4. An important concern I have is whether classifier-free guidance can still be applied on a flow trained with a disentangled source. Does it work out of the box? Or does some adjustment have to be made? (Such as all classes sampling from a common N(0, I) with some probability p.)
    - How does the proposed method compare with existing approaches when CFG is applied? Since many state-of-the-art results with flow models are achieved by applying guidance.

I am highly amenable to improving my score if my concerns are addressed.

[w-a] Aram-Alexandre Pooladian, Heli Ben-Hamu, Carles Domingo-Enrich, Brandon Amos, Yaron Lipman, and Ricky TQ Chen. Multisample flow matching: Straightening flows with minibatch couplings. 2023.

[w-b] Alexander Tong, Nikolay Malkin, Guillaume Huguet, Yanlei Zhang, Jarrid Rector-Brooks, Kilian Fatras, Guy Wolf, and Yoshua Bengio. Improving and generalizing flow-based generative models with minibatch optimal transport, 2023.

[w-c] Xingchao Liu, Chengyue Gong, and Qiang Liu. Flow straight and fast: Learning to generate and transfer data with rectified flow. 2022.

**Questions:**

## Questions & Suggestions
- In Figure 6 caption, L446, do you mean DDPM isn't shown on the **RHS**, not LHS? I think it is a bit strange to have something in the legend and not on the graph. Consider changing the axis to a logarithmic scale, or simply removing DDPM from the legend entirely.
- To that end, I am not sure why the baseline DDPM of Ho et al. [q-a], an SDE was included in a flow matching paper for comparison at all, that too in terms of NFE. It is well established that DDPM usually takes ~1000 NFEs. A better comparison would have been some probability flow ODE version of a diffusion model, typically the state-of-the-art EDM/EDM2, by Karras et al. [q-b]
- In Fig. 9, please show the captions associated with the images, otherwise it is difficult to evaluate how faithful the results are. From visual inspection alone, it is not possible to say which one is better.


[q-a] Jonathan Ho, Ajay Jain, and Pieter Abbeel. Denoising diffusion probabilistic models, 2020

[q-b] https://github.com/NVlabs/edm2

---

### Official Review · Reviewer_U5AY · 2024-11-04

**Soundness:** 3
**Presentation:** 3
**Contribution:** 2
**Rating:** 5
**Confidence:** 3

**Summary:**

The authors presents a novel method for learning conditional flow-matching generative models by matching the flow for conditional priors instead of an unconditional prior that is shared for all classes. Each of the conditional priors are taken to be a Gaussian around the class conditional distribution in the data space, leading to shorter path between prior samples and data samples.
The methods boosts generated samples quality, and shows superior performance with limited NFE, when compared to the baselines.

**Strengths:**

* The proposed method offers a simple extension to flow-matching that is easy to implement and seems to lead to improved image quality.
* The paper is well written and presents the method clearly and concisely.

**Weaknesses:**

* The novelty is rather limited, since the proposed conditional prior is assumed to be a Gaussian. As a result, it is not clear if the proposed method can be effective for more complex image categories, or as a general conditional PDF estimator for other modalities. (i.e., it is easy to construct a data distribution where the class conditional Gaussians are identical for multiple classes). Can you discuss potential extensions to handling more complex conditional priors, or add an experiment showing the performance on a case where priors overlap?
* The training details are missing form the paper, and so do the details of the pre-trained models that were used (i.e., VQ-VAE).

**Questions:**

* Line 398 Fig. 5: DDPM can 2.92 FID [Improved Denoising Diffusion Probabilistic Models], this plot makes it seems as if DDPM cannot surpass the proposed method due to the selective choice of NFE. Can you add NFE values that allows DDMP to reach peak performance?
* Line 423: “We perform flow matching in the latent representation of a pre-trained auto-encoder” - VQ-VAE has a discrete latent space. How is flow-matching in such a discrete space? What auto-encoder did you use exactly?

---

### Note · Authors · 2024-11-14

**Comment:**

We sincerely thank the reviewers, ACs, and PCs for their invaluable feedback and insights on our paper. After careful consideration, we have decided to withdraw the paper to refine it based on the constructive comments. Thank you once again for your time and thoughtful evaluation.

**Withdrawal Confirmation:**

I have read and agree with the venue's withdrawal policy on behalf of myself and my co-authors.